# Leveraging machine learning to unravel the impact of cadmium stress on goji berry micropropagation

**Musab A. Isak**[1], **Taner Bozkurt**[2], **Mehmet Tütüncü**[3], **Dicle Dönmez**[4], **Tolga İzgü**[5], **Özhan Şimşek**[1,6] *

**1** Department of Agricultural Science and Technology, Graduate School of Natural and Applied Sciences Erciyes University, Kayseri, Türkiye, **2** Tekfen Agricultural Research Production and Marketing Inc., Adana, Türkiye, **3** Department of Horticulture, Faculty of Agriculture, Ondokuz Mayıs University, Samsun, Türkiye, **4** Biotechnology Research and Application Center, Çukurova University, Adana, Türkiye, **5** Institute of BioEconomy, National Research Council of Italy (CNR), Florence, Italy, **6** Department of Horticulture, Faculty of Agriculture, Erciyes University, Kayseri, Türkiye

* ozhan12@gmail.com

**Data Availability Statement:** All relevant data are within the manuscript and its Supporting Information files.

## Abstract

This study investigates the influence of cadmium (Cd) stress on the micropropagation of Goji Berry (*Lycium barbarum* L.) across three distinct genotypes (ERU, NQ1, NQ7), employing an array of machine learning (ML) algorithms, including Multilayer Perceptron (MLP), Support Vector Machines (SVM), Random Forest (RF), Gaussian Process (GP), and Extreme Gradient Boosting (XGBoost). The primary motivation is to elucidate genotype-specific responses to Cd stress, which poses significant challenges to agricultural productivity and food safety due to its toxicity. By analyzing the impacts of varying Cd concentrations on plant growth parameters such as proliferation, shoot and root lengths, and root numbers, we aim to develop predictive models that can optimize plant growth under adverse conditions. The ML models revealed complex relationships between Cd exposure and plant physiological changes, with MLP and RF models showing remarkable prediction accuracy ($R^2$ values up to 0.98). Our findings contribute to understanding plant responses to heavy metal stress and offer practical applications in mitigating such stress in plants, demonstrating the potential of ML approaches in advancing plant tissue culture research and sustainable agricultural practices.

## Introduction

Goji berry (*Lycium barbarum*, L.) is a species that belongs to the Solanaceae family and is native to Asia but can also be grown in tropical and subtropical regions of the world [1]. The Goji berry species used in the project (*L. barbarum* L.; 2n = 2x = 24) is diploid with 24 chromosomes [2]. *L. barbarum* L., *L. chinense* M., and *L. ruthenicum* M. are the most commonly used goji berries worldwide, with 89 different species grown in various regions. Goji berry, or wolfberry, is a shrub-like plant with red, soft fruit traditionally used in medicine and food [3].

**Funding:** The author(s) received no specific funding for this work.

**Competing interests:** The authors have declared that no competing interests exist.

While it is found in tropical regions, it primarily grows in dry, semi-dry, slightly salty areas [4]. Goji berry plants are generally spiny shrubs that can grow up to 1–4 meters tall [5]. Goji berry is not selective in location, but it can tolerate dry summer days and grows well in light-sandy, medium-strength, and heavy-clay soil types [6]. Goji berries are typically consumed as dried fruit and add flavor and color to chocolate, pastries, desserts, salads, and granola. Due to their anti-aging, metabolism-boosting, skin-renewing, and cell-growth-accelerating properties, they have also been used in many cosmetic products to increase skin vitality and elasticity, prevent sunspots, and reduce scars and blemishes [7, 8]. Goji berries are known for their high antioxidant and vitamin content, making them effective against various health issues. Goji berry is a deciduous, shrubby plant that is tolerant to cold and hot weather and can grow in almost any type of soil. Its fruits and leaves are rich in chemical compounds such as polysaccharides, fatty acid compositions, carotenoid accumulations, and mineral contents, which make it a valuable medicinal and aromatic plant. It has been used in Chinese traditional medicine for over 2000 years due to its potential health benefits and traditional uses [9].

The increasing demand for goji berry plants has led to the development of micropropagation techniques for mass production. Micropropagation is when small plant parts regenerate whole plants under aseptic conditions on *in vitro*. This technique offers several advantages over traditional propagation methods, including rapid multiplication, disease-free plant production, and year-round availability of planting material. Micropropagation of goji berry typically involves the selection of a suitable explant, surface sterilization, and culture initiation on an appropriate growth medium. The process can be optimized by adjusting hormone concentrations, light intensity, and temperature. The resulting plantlets are then acclimatized to the soil and transplanted into the field. Several studies have reported successful micropropagation of goji berry using different explants and growth media. For example, shoot tips have been used as explants to produce high-quality plantlets using MS medium supplemented with plant growth regulators such as BA and NAA [10]. In another study, nodal segments produced multiple shoots on an MS medium supplemented with cytokinins and auxins [11]. Overall, micropropagation is a promising technique for the mass production of high-quality goji berry plants. It can help meet the increasing demand for goji berries and contribute to the development of the goji berry industry.

The accumulation of Cd in the soil and plants because of Cd contamination has been an important research topic in recent years due to its potential impact on human health [12, 13]. In addition to natural Cd in the soil, Cd mainly enters the soil through anthropogenic sources such as atmospheric deposition, sewage sludge application to agricultural lands, and fertilizers [14–16]. Cd can accumulate in plant tissues at toxic levels, limiting plant growth. Although not an essential nutrient for plants, it can easily be taken up by plant roots and accumulate in plant products at levels that pose a risk to food chains. When present in high concentrations in plants, animals, and humans, Cd is a toxic element that can cause health problems such as lung, liver, and kidney diseases, vision impairment, anemia, and high blood pressure [14, 17, 18].

Cd is a toxic heavy metal that can accumulate in plant tissues and negatively affect plant growth and development. Several studies have investigated the impact of Cd on different plant species [12].

*In vitro* techniques can be used to study the effects of Cd stress on plant growth and development. In this process, plants are grown under controlled conditions in a sterile environment with different concentrations of Cd in the growth medium. Here are some studies investigating the effects of *in vitro* Cd stress on plant growth and development. Zhang et al. [19] studied the effects of Cd stress on the growth and physiological characteristics of potato (*Solanum tuberosum* L.) plantlets on *in vitro*. They found that exposure to Cd caused a decrease in plant growth, photosynthesis, and chlorophyll content. Kaur and Bhandari [20] investigated the

effect of Cd stress on the *in vitro* growth and antioxidant defence system of a medicinal plant, *Phyllanthus amarus* Schum. & Thonn. They found that Cd stress caused a significant decrease in plant growth, biomass, and antioxidant enzymes, such as superoxide dismutase (SOD), catalase (CAT), and peroxidase (POD). Rahmati and Ghasemnezhad [21] studied the effect of Cd stress on *in vitro* growth and accumulation of Cd in two cultivars of *Rosa sp*. Overall, these studies suggest that *in vitro* Cd stress can negatively affect plant growth and development and physiological and biochemical processes in plants.

The accumulation of Cd in the soil and its plant uptake pose significant challenges to agricultural productivity and food safety. Cd induces various physiological and biochemical changes in plants, leading to growth inhibition, oxidative stress, and disruptions in photosynthesis and nutrient uptake. Traditional methods of studying these effects often involve labor-intensive experiments and complex data analysis, which can be time-consuming and may not fully capture the intricate interactions within the plant system [22]. This is where machine learning (ML) comes into play. ML techniques can potentially revolutionize how we analyze and interpret complex biological data. By leveraging ML algorithms, we can process large datasets more efficiently, identify patterns and relationships that may not be immediately apparent, and develop predictive models to forecast plant responses under different environmental conditions. In plant tissue culture, ML can optimize experimental conditions, improve the accuracy of predictive models, and ultimately enhance our understanding of how plants respond to stress factors such as Cd.

ML is a prominent subfield in artificial intelligence that can predict and classify outcomes based on specific inputs [23, 24]. By utilizing ML techniques, computers can autonomously learn and convert data into meaningful knowledge, thus eliminating the need for explicit human programming [25–27].

However, despite its popularity in various scientific domains, the application of ML in plant tissue culture studies is relatively limited [28]. In the context of *in vitro* micropropagation, a complex biological process influenced by genotypes, culture media, and conditions, traditional statistical methods often struggle to analyze large and intricate datasets [29]. Nevertheless, recent advancements have demonstrated the successful application of ML models, such as neural networks and decision-tree-based models, in predicting and optimizing plant tissue culture procedures [30, 31].

Numerous ML models effectively forecast and refine plant tissue culture procedures across various studies [32–37]. ML has demonstrated significant potential in enhancing various aspects of *in vitro* culture systems, providing improvements over traditional statistical methods. Conventional approaches often struggle with the complexity and non-linearity of biological processes in plant tissue culture. In contrast, ML algorithms, such as artificial neural networks (ANNs), support vector machines (SVMs), and genetic algorithms (GAs), offer higher accuracy and efficiency. For instance, ML has been successfully applied to optimize sterilization protocols [38, 39], seed germination conditions [31, 40], and callus induction processes [41–44]. Additionally, ML has optimized somatic embryogenesis [45, 46] and haploid production [47, 48]. Furthermore, ML has been used to improve gene transformation efficiency [49, 50], indirect shoot regeneration [51], root formation [52, 53] and, secondary metabolite production [54], shoot proliferation [45, 46, 55–57]; these applications highlight the versatility and effectiveness of ML in improving *in vitro* culture outcomes. By leveraging ML techniques, this study aims to enhance the precision and predictive power in assessing genotype-specific responses to cd stress in goji berry micropropagation, thereby providing valuable insights for developing more resilient crop varieties. This study utilized five different ML models: multilayer perceptron (MLP), random forest (RF), Gaussian process (GP), support vector machines (SVM), and extreme gradient boosting (XGBoost), each with its unique

strengths and capacity to capture complex relationships within the data. By applying these models, the study aimed to understand better the intricate relationships between goji berry genotypes, micropropagation, and rooting efficiency. Using diverse ML techniques reflects a deliberate effort to enhance the study's ability to analyze the complex dataset. The primary objective of this study is to elucidate the effects of Cd stress on the micropropagation of different goji berry genotypes using advanced ML algorithms. By integrating the topics of goji berry cultivation, Cd stress, and ML, this research aims to uncover genotype-specific responses to heavy metal stress and develop predictive models that can optimize plant growth under adverse conditions. The convergence of these areas addresses the critical need for innovative approaches in agricultural practices, particularly in enhancing the resilience of crops to environmental contaminants. This study contributes to understanding plant responses to Cd stress and demonstrates the potential of ML techniques to revolutionize plant tissue culture and sustainable agricultural practices.

## Material and methods

### Plant material

As plant material, NQ1 and NQ2 commercial goji berry varieties and a source of unknown origin grown at the Campus of Erciyes University, Kayseri, Türkiye (ERU) were used. Commercial varieties were obtained from a specialized nursery company.

### Nutrient medium and culture of shoot tips

The shoot tips of the plant material used in the study were washed under tap water for ten minutes, followed by soaking in 70% ethanol for 3 minutes and a 20% sodium hypochlorite (NaClO) solution for 10 minutes. Afterwards, they were washed three times with sterile distilled water. After the sterilization stage, the shoot tips were cultured in MS (Murashige and Skoog, 1962) nutrient medium containing 1 mg/L BA, 30 g/L sucrose, and 8 g/L agar. The cultures were incubated under a 16-hour light/8-hour dark photoperiod and 25°C temperature conditions.

### Micropropagation and rooting under *in vitro* Cd stress

The Cd stress experiment was conducted with the cultured and propagated plants. For this purpose, MS nutrient media containing 0, 100, 200, 300, 400, and 500 μM concentrations of Cd were prepared, and the plants were transferred to these media supported with 1 mg/L BAP. The plants were cultured under 16-hour light/8-hour dark and 25°C temperature conditions. The plants were subcultured every 4 weeks 3 times. At the end of each subculture, the micropropagation coefficient (number of plantlets/plant) and plant height (cm) were calculated. To investigate *in vitro* rooting under Cd stress conditions, 0, 100, 200, 300, 400, and 500 μM concentrations of Cd were added to MS media containing 1 mg/L IBA, and plants from Cd-free control media were transferred to rooting media. After 8 weeks, rooting rate (%), root length (cm), and number of roots per plant were calculated.

### Statistical analysis

The micropropagation and rooting of NQ7, NQ1, and ERU Goji berry cultivars under *in vitro* conditions of Cd stress were examined using a factorial order random plot trial design. The data obtained was analyzed using an analysis of variance (ANOVA), and the software programs exhibiting significant changes were subjected to the LSD test. The statistical analyses

were performed using the R-programming language. The Pearson correlation coefficients for the parameters were computed using the corrplot package in R software version 4.3.1.

## Modeling procedure

The aim was to estimate the output (proliferation, shoot length, number of roots, and root length) variables using this input (different concentrations of 0 μM, 100 μM, 200 μM, 300 μM, 400 μM, and 500 μM and three different genotypes) for modeling. In this study, the dataset was divided into training and testing subsets using a 10-fold cross-validation method. Specifically, the dataset was randomly partitioned into 10 equal parts. Each part was used as a testing set once while the remaining nine parts were used as the training set. This process was repeated 10 times to ensure each part of the data was used for testing exactly once, providing a robust evaluation of the model's predictive performance. To ensure the partitions were representative, stratification was employed, meaning that the distribution of the target variable was consistent across all folds. Additionally, the overall ratio of the dataset split into training and testing was maintained at 80% for training and 20% for testing in each fold. This stratified approach helps to maintain the balance of class labels in each fold, thereby improving the predictive performance of the MLP and ML models (Fig 1A). R-programming was used to implement coding with the help of Caret packages. The Caret package was used for data partitioning, model training, prediction, and performance evaluation. Specifically, we used the train function for training the models, specifying the method parameter for each algorithm (e.g., method = 'rf' for Random Forest, method = 'svmRadial' for Support Vector Machines with radial basis function kernel). The trainControl function was employed to set up the cross-validation method (method = 'cv', number = 10) and other relevant parameters. Several metrics were used to assess and compare the accuracy and precision of MLP and ML models. These metrics

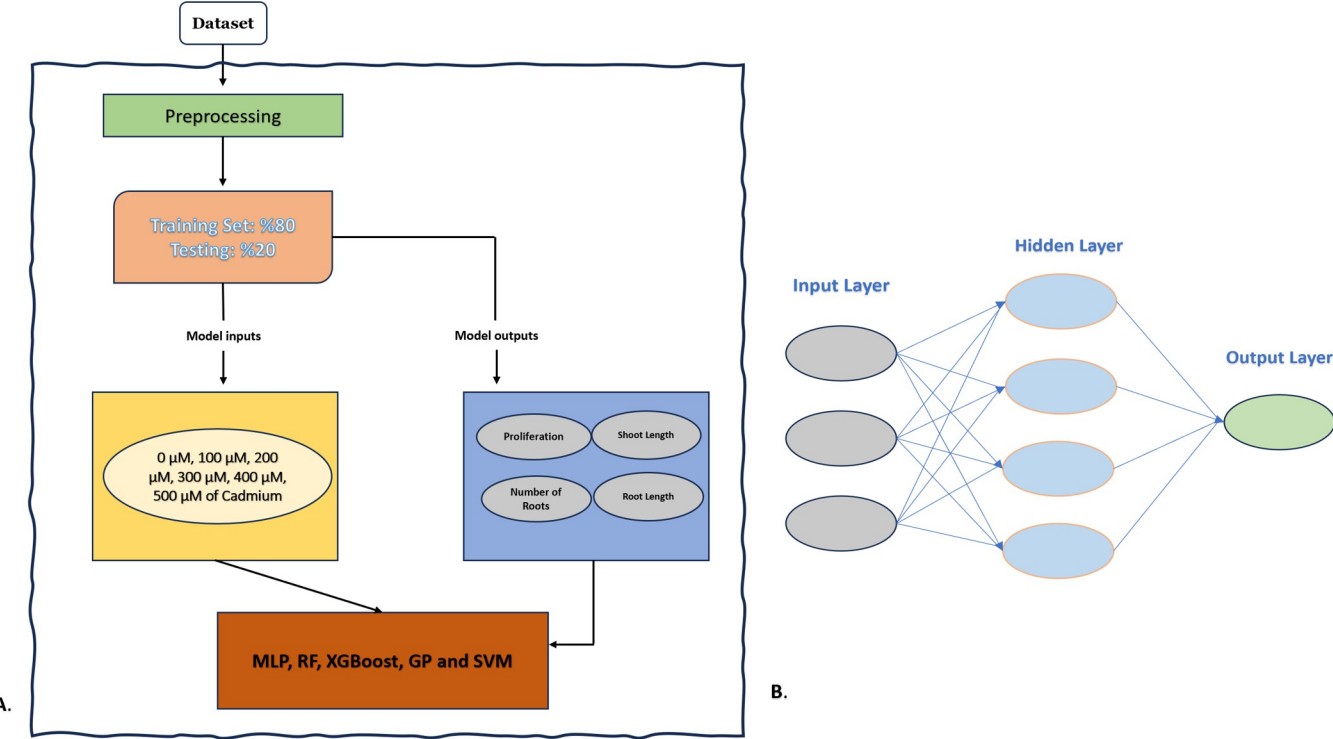

**Fig 1.** A. Schematic diagram of step-by-step ML approach of this study, B. Multilayer perceptron.

included the coefficient of determination ($R^2$), which shows the degree of relationship between the model and the dependent variable Eq 1; and Root Mean Square Error (RMSE), which shows how closely the regression line matches the observed data points in Eq 2, and the Mean Absolute Error (MAE), which calculates the average error between the predicted and observed values in Eq 3.

$$R^2 = 1 - \frac{\sum_{i=1}^{n} \left(Y_i - \hat{Y}_i\right)^2}{\sum_{i=1}^{n} \left(Y_i - \tilde{Y}\right)^2} \tag{1}$$

$$RMSE = \sqrt{\frac{\left(\sum_{i=1}^{n} \left(Y_i - \hat{Y}_i\right)^2\right)}{n}} \tag{2}$$

$$MAE = \frac{1}{n} \sum_{i=1}^{n} |Y_i - \hat{Y}_i| \tag{3}$$

While $Y_i$ = actual value, $\hat{Y}_i$ = Predicted value, $\tilde{Y}$ = mean o the actual values and n = sample count.

**Multilayer perceptron.** A multilayer perceptron (MLP) is a feedforward neural network consisting of a series of processing nodes arranged in a linear, feedforward configuration. It consists of multiple layers, all of which are fully interconnected. The MLP has an input layer, an output layer, and one or more hidden layers (Fig 1B). Using backpropagation in the training process involves adjusting the weights and biases by minimizing the error expressed in Eq 4 until the desired accuracy is achieved.

$$E = \frac{1}{n} \sum_{n=1}^{n} (y_s - \hat{y}_s) \tag{4}$$

y: the observed value of data point n; n: number of samples.

**Support Vector Machines.** Support Vector Machines (SVMs) are a class of algorithms that encompass both supervised and unsupervised learning techniques. These algorithms are designed for regression analysis, clustering, and classification tasks. Two common types of SVMs are Support Vector Classification (SVC) and Support Vector Regression (SVR). One of the advantages of SVMs is that they are efficient even with small datasets. Additionally, SVMs address issues that conventional AI algorithms often struggle with, such as overfitting, low convergence rates, and local minima entrapment. The SVM algorithm is defined by Eq 5, which helps determine the optimal separator plane for distinguishing between classes.

$$f(x) = w\varphi(x) + b \tag{5}$$

**Random Forest.** Random Forest (RF), a technique for ensemble learning comprised of unpruned trees that was innovated by Breiman [58], has demonstrated effectiveness in both regression and classification tasks. Due to its efficiency and ease of implementation, RF can prevent overfitting, effectively manage noise, and handle many features. The RF model incorporates randomness by introducing two sources into each tree during construction, reducing correlation while preserving individual strengths. During training, bootstrap replicas are employed, and random variable subsets are used to create optimal splits. The trees that have reached maturity, characterized by low bias and high variance, are utilized for both regression

and classification tasks, as specified in Eq 6.

$$y = \sum_{i=1}^{n}(a_i - a_i^*)k(x, x_i) + b \qquad (6)$$

y = data point value; n = sampling size (number). $ai$ and $a_i^* a_i^*$ are coefficients associated with the predictor variables, $k(x,x_i)$ computes the similarity between the input vector $x$ and the $i$-th data point $x_i x_i$. $b$ is the bias term or intercept

**Gaussian Process.** The Gaussian Process (GP) model, a widely used supervised learning technique, extends the Gaussian probability distribution to encapsulate the spread of random variables. Applicable for both classification and regression tasks, the GP model determines the probability of input samples belonging to particular classes. This method is particularly advantageous for small datasets as it provides consistency, accuracy, and computational efficiency. Eq 7 illustrates the derivation process, which relates the input (x) to the output (y) for each individual data point.

$$y_i = f(x_i) + \varepsilon \qquad (7)$$

**Extreme Gradient Boost.** The XGBoost algorithm is a highly effective tool for addressing regression and classification challenges. It is a member of the gradient boosting decision tree family and is distinguished by its exceptional performance and rapid processing capabilities. In a gradient boosting framework, XGBoost excels at learning from errors and progressively reducing the error rate through multiple iterations.

## Results

This study explores the repercussions of different Cd concentrations on micropropagation of three distinct genotypes: ERU, NQ1, and NQ7. Cd, recognized as a hazardous heavy metal, poses potential threats to living organisms, particularly in the context of ecological systems. The experimental design involved subjecting the genotypes mentioned above to a range of Cd concentrations (0 μM, 100 μM, 200 μM, 300 μM, 400 μM, and 500 μM), with subsequent measurement of their multiplication rates (proliferation).

The results indicate discernible variations in the multiplication rates across genotypes and Cd concentrations. For all genotypes, a gradual decline in multiplication rates is observed with increasing Cd concentrations. Specifically, ERU exhibited the highest multiplication rate at the absence of Cd at a rate of (7.63). At the same time, ERU with Cd 500 μM showed the lowest micropropagation rate, gradually decreasing as concentrations escalated. Similarly, NQ1 and NQ7 consistently reduced multiplication rates with elevated Cd levels Table 1. Notably, the multiplication rates of all genotypes experienced a substantial decline at higher Cd concentrations, reflecting this heavy metal's deleterious impact on the studied organisms' reproductive

**Table 1. Effect of Cd concentrations on the proliferation of different genotypes.**

|  | Cd (0 μM) | Cd (100 μM) | Cd (200 μM) | Cd (300 μM) | Cd (400 μM) | Cd (500 μM) | Genotype Average |
|---|---|---|---|---|---|---|---|
| **ERU** | 7.63±0.66 | 7.23±0.23 | 6.82±0.45 | 5.54±0.41 | 3.55±0.41 | 2.56±0.67 | 5.56±2.10 |
| **NQ1** | 7.41±1.36 | 6.72±0.55 | 6.37±0.49 | 5.85±0.84 | 3.43±0.67 | 2.13±0.39 | 5.32±2.07 |
| **NQ7** | 7.18±0.65 | 6.93±0.32 | 6.50±0.34 | 5.52±0.67 | 3.41±0.33 | 2.10±0.33 | 5.27±2.05 |
| **Cd Average** | 7.41±0.17A | 6.96±0.25B | 6.56±0.18 B | 5.63±0.18 C | 3.46±0.06 D | 2.26±0.24 E |  |

LSD $_{Genotype}$: N.S, LSD $_{Cd\ Concentration}$: 0.412***, LSD $_{Genotype^*\ Cd\ Concentration}$: N.S.

N.S.: Not significant.

**Table 2. Effect of Cd concentrations on shoot length of different genotypes.**

| | Cd (0 µM) | Cd (100 µM) | Cd (200 µM) | Cd (300 µM) | Cd (400 µM) | Cd (500 µM) | Genotype Average |
|---|---|---|---|---|---|---|---|
| ERU | 5.24±0.79 | 5.13±0.50 | 4.01±0.58 | 3.17±0.37 | 2.31±0.46 | 1.74±0.22 | 3.60±1.29 |
| NQ1 | 5.65±0.65 | 5.08±0.52 | 4.29±0.70 | 3.22±0.64 | 2.63±0.33 | 1.84±0.28 | 3.78±1.33 |
| NQ7 | 5.56±0.81 | 5.06±0.46 | 4.17±0.53 | 3.20±0.46 | 2.75±0.34 | 2.03±0.14 | 3.79±1.33 |
| Cd Average | 5.48±0.20A | 5.09±0.03B | 4.15±0.11 C | 3.19±0.02D | 2.56±0.18E | 1.87±0.11F | |

LSD $_{Genotype}$: N.S, LSD $_{Cd\ Concentration}$: 0.268***, LSD $_{Genotype*\ Cd\ Concentration}$: N.S.

N.S.: Not significant.

capacities. The averaged multiplication rates across genotypes reinforce the overall trend, illustrating a Cd concentration-dependent decrease in multiplication rates.

The study also delves into the impact of varying Cd concentrations on shoot length across the mentioned genotypes. Table 2. reveals discernible differences in shoot lengths across genotypes and Cd concentrations. ERU exhibited a consistent reduction in shoot length as Cd concentrations increased, reflecting its heightened sensitivity to Cd-induced stress. Similarly, NQ1 and NQ7 gradually declined shoot length with escalating Cd concentrations. Notably, at higher Cd concentrations, a substantial reduction in shoot length is observed across all genotypes, emphasizing the adverse effects of Cd on plant growth. Overall, NQ1 without Cd showed the highest shoot length, while ERU at 500 µM showed the lowest.

The averaged shoot lengths across genotypes further underscore the concentration-dependent inhibition of growth induced by Cd.

The results of the root length study, as presented in Table 3, demonstrate that genotype NQ7 showed the greatest root length in the absence of Cd, with a measurement of 4.83. However, when exposed to 500 µM Cd, NQ7 displayed no root length. It is worth noting that under Cd-free conditions, all three genotypes exhibited considerable root lengths. Furthermore, increased Cd concentration resulted in a progressive decrease in root length for all genotypes. These findings emphasize the influence of Cd concentration on the development of roots in the examined genotypes.

Table 4 presents data on the influence of varying Cd concentrations (0 µM, 100 µM, 200 µM, 300 µM, 400 µM, 500 µM) on the root number of three different genotypes (ERU, NQ1, and NQ7). Across all genotypes, an increase in Cd concentration was associated with a consistent decrease in root number. While NQ7 consistently exhibited the highest number of roots at a rate of 4.8, it also experienced a considerable reduction in root growth as Cd concentrations increased. Notably, at the highest Cd concentration (500 µM), all genotypes experienced a complete cessation of root growth. The overall Cd averages support the descending

**Table 3. Effect of Cd concentrations on root length of different genotypes.**

| | Cd (0 µM) | Cd (100 µM) | Cd (200 µM) | Cd (300 µM) | Cd (400 µM) | Cd (500 µM) | Genotype Average |
|---|---|---|---|---|---|---|---|
| ERU | 3.75±2.08 | 3.19±2.23 | 1.75±1.87 | 0.70±1.13 | 0.33±0.70 | 0.00±0.0 | 1.62±1.52 B |
| NQ1 | 4.18±1.51 | 3.40±1.80 | 3.04±1.62 | 2.07±1.79 | 1.27±1.64 | 0.25±1.79 | 2.36±1.38 A |
| NQ7 | 4.83±0.59 | 3.93±1.41 | 2.60±1.80 | 1.10±1.47 | 0.65±1.06 | 0.00±0.0 | 2.18 ±1.50 A |
| Cd Average | 4.25±0.54 A | 3.50±0.37B | 2.46+0.65C | 1.29±0.66D | 0.75±0.46DE | 0.08±0.12E | |

LSD $_{Genotype}$: 0.515**, LSD $_{Cd\ Concentration}$: 0.729***, LSD $_{Genotype*\ Cd\ Concentration}$: N.S.

N.S.: Not significant

**Table 4. Effect of Cd concentrations on number of roots of different genotypes.**

|  | Cd (0 μM) | Cd (100 μM) | Cd (200 μM) | Cd (300 μM) | Cd (400 μM) | Cd (500 μM) | Genotype Average |
|---|---|---|---|---|---|---|---|
| ERU | 3.9±2.18 | 2.7±2.00 | 1.6±1.78 | 0.9±1.44 | 0.5±1.08 | 0.0±0.0 | 1.60±1.61B |
| NQ1 | 4.3±1.64 | 3.0±1.70 | 2.7±1.49 | 1.9±1.66 | 1.1±1.45 | 0.2±0.63 | 2.20±1.42 A |
| NQ7 | 4.8±0.79 | 4.4±1.65 | 2.7±2.00 | 1.2±1.62 | 1.0±1.63 | 0.0±0.0 | 2.35±1.71 A |
| Cd Average | 4.33±0.45A | 3.36±0.71B | 2.33±0.63C | 1.33±0.52D | 0.86±0.30D | 0.06±0.09E |  |

LSD $_{Genotype}$: 0.534*, LSD $_{Cd\ Concentration}$: 0.755***, LSD $_{Genotype*\ Cd\ Concentration}$: N.S.

N.S.: Not significant.

trend in root length with increasing Cd levels, providing valuable insights into the varying sensitivities of the genotypes to Cd-induced stress.

Table 5 indicates genotype-specific responses to Cd exposure, with 'ERU,' 'NQ1', and 'NQ7' exhibiting distinct rooting rate patterns across the range of Cd concentrations. 'ERU' demonstrates a steady decline in rooting rate as Cd concentrations increase, suggesting heightened sensitivity to Cd-induced stress. In contrast, 'NQ1' and 'NQ7' show a more gradual rooting rate decrease with escalating Cd levels. Overall, 'NQ7' had the highest rate of 100 without Cd. The calculated average rooting rates across genotypes highlight a concentration-dependent reduction in rooting rates, with a clear trend of decreasing averages as Cd concentrations rise.

Pearson correlation coefficients for various physiological parameters in three plant varieties ('NQ1', 'NQ7', and 'ERU') under different concentrations of Cd. Strong positive correlations are observed between the parameters such as proliferation (MR), shoot height (SH), rooting rate (RR %), root length (RL in cm), and number of roots (NR) (Fig 2).

## Machine learning analysis

Machine learning algorithms employ statistical and computational methods to identify patterns in data, establish models from those patterns, and then utilize those models to make predictions or judgments. In this study, we employed MLP, SVM, RF, GP, and XGBoost algorithms to predict the relationship between inputs and outputs, compare and evaluate the performance of the models, and analyze the data generated from the experiments. The input variables for this study comprised three distinct genotypes and six different Cd concentrations. The output variables included observations derived from *in vitro*-cultivated explants of goji berry plants. RMSE, $R^2$, and MAE were employed as evaluation metrics for the model's validity. The $R^2$ values range from 0 to 1, with 1 representing an ideal prediction and 0 signifying no predictive capacity. Model precision is represented by the RMSE values, which typically range from 0 to positive infinity. Lower values indicate improved performance. Additionally, MAE represents the predicted accuracy and ranges from 0 to positive infinity, with lower

**Table 5. Effect of Cd concentrations on root rate of different genotypes.**

|  | Cd (0 μM) | Cd (100 μM) | Cd (200 μM) | Cd (300 μM) | Cd (400 μM) | Cd (500 μM) | Genotype Average |
|---|---|---|---|---|---|---|---|
| ERU | 80±42.2 | 70±48.3 | 50±52.7 | 30±48.3 | 20±42.1 | 0±0.0 | 41.66±49.7 B |
| NQ1 | 90±31.6 | 80±42.2 | 80±42.2 | 60±51.6 | 40±51.6 | 10±31.6 | 60.00±49.4 A |
| NQ7 | 100±0.0 | 90±31.6 | 70±48.30 | 40±51.6 | 30±48.30 | 0±0.0 | 55.00±50.2 AB |
| Cd Average | 90±14A | 80±17AB | 66.7±11 B | 43±6.6 C | 30±10C | 3.33±3.33 D |  |

LSD $_{Genotype}$: 14.40*, LSD $_{Cd\ Concentration}$: 20.37***, LSD $_{Genotype*\ Cd\ Concentration}$: N.S.

N.S.: Not significant.

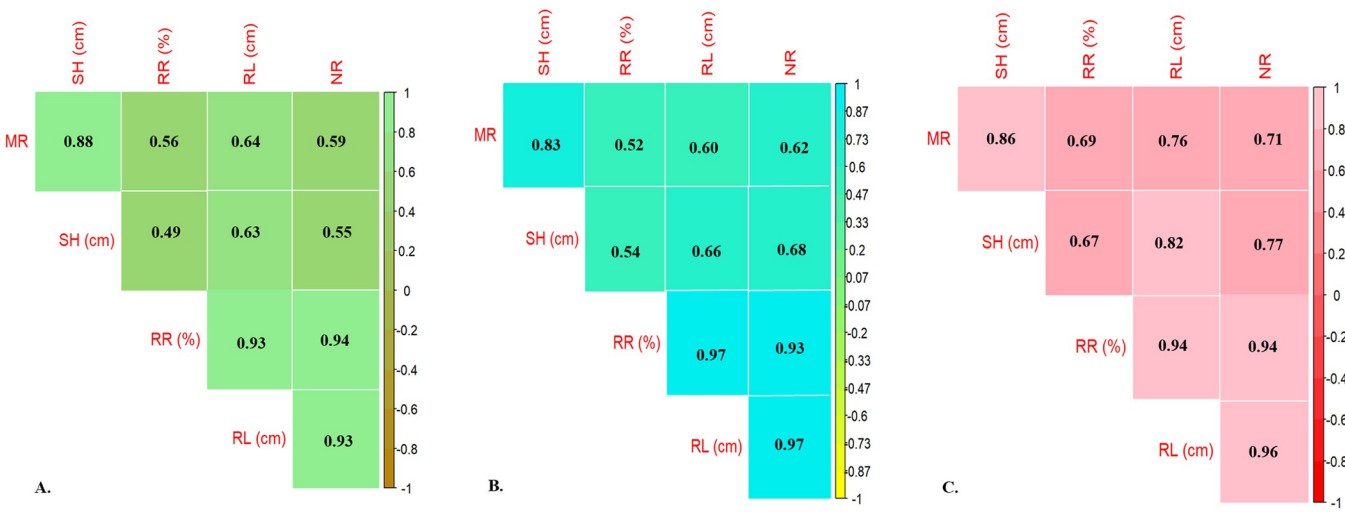

**Fig 2.** Correlation triangle obtained for studied morphological traits of *in vitro* grown seedlings of goji berry varieties: A. ERU variety; B. NQ1 variety; C. NQ7 variety. MR: Multiplication Rate; SH: Shoot Height; RR: Root Rate; RL: Root Length; NR: Number of Roots.

values signifying greater accuracy. Table 6. summarizes the study's results and shows the outputs of the machine learning models used in the research.

Table 6 illustrates a comprehensive assessment of the performance metrics of training and testing data for various machine learning models utilized in predicting plant growth variables,

**Table 6. Performance metrics of ANN-based MLP and ML models.**

| Model | Parameter | RMSE | | MAE | | R² | |
|---|---|---|---|---|---|---|---|
| | | Training | Testing | Training | Testing | Training | Testing |
| MLP | Proliferation | 0.07 | 0.07 | 0.05 | 0.06 | 0.91 | 0.94 |
| | Shoot Length | 0.09 | 0.08 | 0.07 | 0.07 | 0.87 | 0.94 |
| | Root Length | 0.06 | 0.06 | 0.04 | 0.05 | 0.97 | 0.99 |
| | Root Number | 0.10 | 0.08 | 0.07 | 0.06 | 0.92 | 0.89 |
| RF | Proliferation | 0.06 | 0.07 | 0.05 | 0.06 | 0.90 | 0.95 |
| | Shoot Length | 0.10 | 0.09 | 0.08 | 0.07 | 0.84 | 0.93 |
| | Root Length | 0.05 | 0.07 | 0.03 | 0.05 | 0.98 | 0.98 |
| | Root Number | 0.09 | 0.07 | 0.06 | 0.04 | 0.93 | 0.99 |
| XGBoost | Proliferation | 0.07 | 0.05 | 0.05 | 0.04 | 0.90 | 0.91 |
| | Shoot Length | 0.11 | 0.08 | 0.08 | 0.07 | 0.82 | 0.91 |
| | Root Length | 0.05 | 0.05 | 0.04 | 0.04 | 0.97 | 0.98 |
| | Root Number | 0.10 | 0.10 | 0.08 | 0.08 | 0.92 | 0.91 |
| GP | Proliferation | 0.08 | 0.07 | 0.06 | 0.06 | 0.86 | 0.92 |
| | Shoot Length | 0.09 | 0.07 | 0.07 | 0.06 | 0.85 | 0.93 |
| | Root Length | 0.08 | 0.08 | 0.06 | 0.06 | 0.96 | 0.95 |
| | Root Number | 0.10 | 0.08 | 0.08 | 0.06 | 0.92 | 0.95 |
| SVM | Proliferation | 0.08 | 0.08 | 0.07 | 0.07 | 0.87 | 0.93 |
| | Shoot Length | 0.09 | 0.08 | 0.07 | 0.07 | 0.85 | 0.91 |
| | Root Length | 0.07 | 0.08 | 0.05 | 0.06 | 0.94 | 0.94 |
| | Root Number | 0.11 | 0.09 | 0.08 | 0.07 | 0.91 | 0.95 |

ANN: Artificial Neural Network; ML: Machine learning; MLP: Multilayer perceptron; RF: Random Fores; XGBoost: Extreme Gradient Boost; GP: Gaussian Process; SVM: Support Vector Machine; R²: Coefficient of determination; MAE: Mean absolute error; RMSE: Root mean square.

such as proliferation, shoot length, root length, and root number. In the Artificial Neural Network-based Multilayer Perceptron (ANN-MLP) model, the testing metrics closely align with the training metrics, indicating consistent performance. For example, the RMSE for proliferation is 0.07 for both training and testing, and the $R^2$ value slightly increases from 0.91 in training to 0.94 in testing, demonstrating the model's strong generalization capability. Similarly, for shoot length and root length, the testing RMSE values are either equal to or lower than the training values, with high $R^2$ values indicating effective prediction accuracy. The root number prediction shows a slight decrease in $R^2$ from training (0.92) to testing (0.89), but overall, the MLP model maintains high performance across parameters.

RF also demonstrates robust predictive power. For root length, the testing $R^2$ value remains outstanding at 0.98, matching the training $R^2$, and the RMSE values are consistently low. The model shows impressive generalization in predicting shoot length, where the testing $R^2$ (0.93) is higher than the training $R^2$ (0.84). Similarly, for root number, the RF model achieves a near-perfect $R^2$ of 0.99 in testing, surpassing its training $R^2$ of 0.93.XGBoost showed competitive performance, particularly in predicting root length, where the training and testing RMSE values are 0.05 and the $R^2$ values are close to 0.98. For proliferation and shoot length, XGBoost exhibits strong generalization with lower testing RMSE values compared to training and testing $R^2$ values, indicating robust predictive capability. The GP model shows moderate performance with testing $R^2$ values ranging from 0.92 to 0.95, slightly lower than some other models but still indicating good predictive power. The model's RMSE and MAE values are generally low, with root length prediction maintaining a high $R^2$ value of 0.95. Lastly, the SVM model displays strong generalization capabilities, testing RMSE values comparable to training RMSE values across all parameters. The testing $R^2$ values are consistently high, particularly for root length (0.94), demonstrating the model's reliable performance in predicting plant growth metrics.

The testing metrics confirm that the machine learning models effectively generalize to unseen data. The MLP and RF models stand out with their low errors and high $R^2$ values, indicating their robustness in predicting plant growth parameters. XGBoost and SVM also show competitive performance, with SVM excelling in root length prediction. While the GP model shows slightly higher errors, it still provides reliable predictions, confirming its utility in this context. These findings highlight the effectiveness of machine learning models in handling complex biological data and improving tissue culture outcomes.

Fig 3 compares the actual and predicted values, with the individual samples depicted on the horizontal axis and the corresponding predicted outcomes displayed on the vertical axis. This visual representation clearly demonstrates the alignment or deviation between the actual and predicted values for the specified dataset.

## Discussion

The recent study on the effects of Cd on micropropagation of three distinct genotypes, ERU, NQ1, and NQ7, highlights the significant impact of Cd stress on plant physiology, including multiplication rates, shoot and root lengths, and root numbers. These results are discussed within the context of existing research, facilitating a comprehensive understanding of plant responses to Cd and strategies for mitigation. Cd's detrimental effects on plant growth and development have been well-documented across various species and genotypes. The observed decline in multiplication rates and inhibition of shoot and root growth with increasing Cd concentrations aligns with findings from Sanità di Toppi et al. [59], who reported similar stress-induced alterations in *in vitro* plants and cell suspension cultures exposed to Cd, including the induction of phytochelatins without lipid peroxidation in carrot plants. This supports

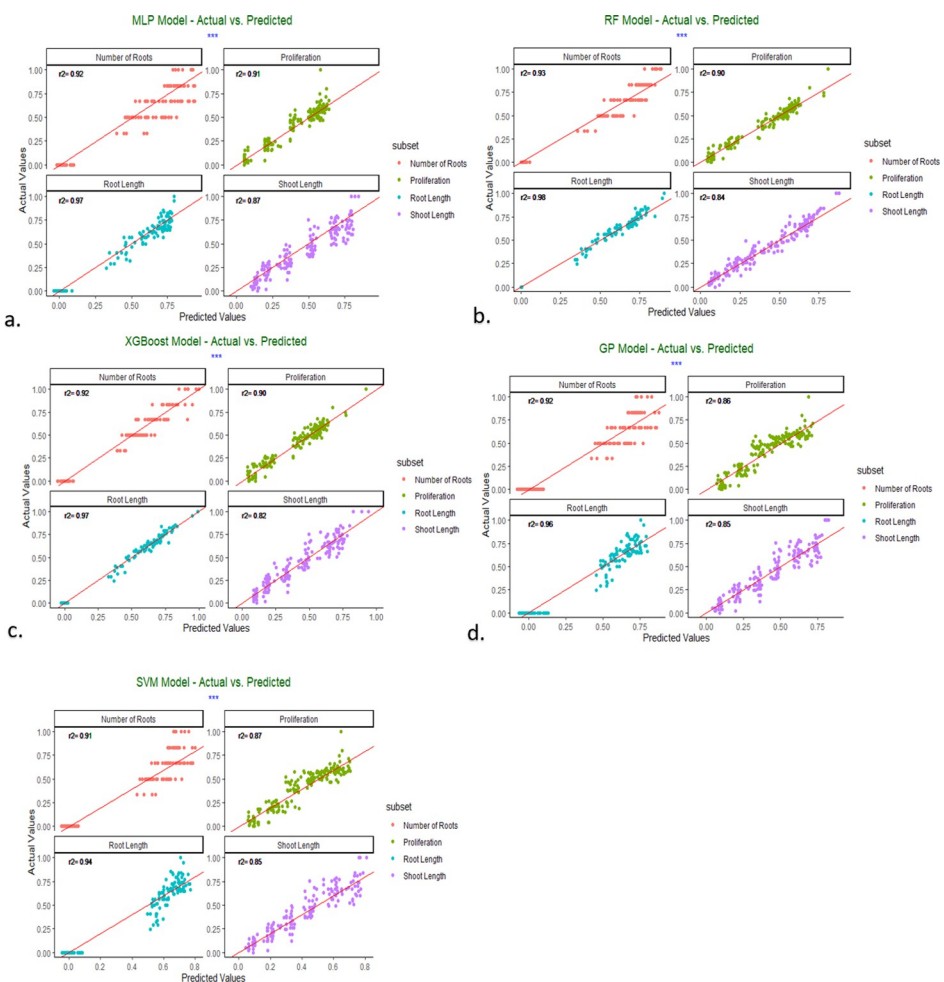

**Fig 3.** Actual and predicted values of ML models, a. MLP model; b. RF model; c. XGBoost model; d. GP model; e. SVM model.

the notion that plants employ specific biochemical pathways, such as phytochelatin synthesis, to manage heavy metal stress, thereby mitigating Cd's toxic effects without directly causing oxidative damage. The variation in Cd-induced responses among different genotypes suggests inherent differences in metal tolerance and accumulation capabilities. This is consistent with the broader literature, indicating species-specific and genotype-specific heavy metal detoxification and tolerance mechanisms. For instance, research by Soleimani et al. [60] on *Narcissus tazetta* revealed a significant accumulation of Cd and alterations in alkaloid content under Cd stress, underscoring the complex interplay between metal stress and secondary metabolite production. Moreover, the increase in peroxidase activity and the appearance of new isoenzymes under Cd stress, as observed in *Narcissus tazetta*, highlight the role of antioxidant defense systems in coping with heavy metal-induced oxidative stress. This adaptive response is pivotal for maintaining cellular homeostasis and protecting plant tissues from Cd-induced damage. Comparatively, the resilience of certain genotypes to Cd, as evidenced by lesser reductions in root and shoot lengths under high Cd concentrations, may offer insights into breeding and biotechnological approaches for enhancing plant tolerance to heavy metals. Research on chickpea cultivars exposed to Cd stress demonstrated genotype-specific differences in growth responses, reinforcing the potential for selecting and engineering crops with enhanced Cd tolerance. The

comprehensive analysis of plant responses to Cd across different studies emphasizes the multifaceted nature of plant-metal interactions involving physiological, biochemical, and molecular adjustments. Future research should focus on elucidating the genetic and molecular bases of Cd tolerance and accumulation, leveraging advances in genomics and biotechnology to develop plants capable of efficient Cd detoxification and phytoremediation [61].

The findings of Baktemur [62] on the tolerance of garlic (*Allium sativum* L.) to various heavy metals, including Cd, under *in vitro* conditions offer a parallel to our observations on the impact of Cd on the micropropagation of three distinct genotypes: ERU, NQ1, and NQ7. Similar to Baktemur's [62] results, where an increase in heavy metal concentrations led to a decline in leaf and root development, with no root formation at the highest concentrations of Cd and other metals, our study also documented a Cd concentration-dependent decrease in multiplication rates, shoot and root lengths, and rooting rates across all genotypes studied. This confluence of findings underscores the broad spectrum of plant responses to heavy metal stress, highlighting Cd's particularly inhibitory effect on plant growth and development. The absence of root formation at high Cd concentrations noted by Baktemur [62] reinforces the critical impact of this metal on root development. This observation aligns with our results showing a significant reduction in root length and number under increasing Cd stress. These parallel observations emphasize the importance of further research, both *in vitro* and *in vivo*, to elucidate the mechanisms of plant tolerance and adaptation to heavy metal stress, ultimately guiding the development of strategies to mitigate these adverse effects. The investigation by Karmous et al. [63] into the ameliorative effects of Zinc Oxide nanoparticles (ZnONPs) on Cd toxicity in *Capsicum annuum* L. presents a novel approach to mitigating heavy metal stress in plants, which complements and broadens the scope of our study on the impact of Cd on the micropropagation of ERU, NQ1, and NQ7 genotypes. While our research documented the inhibitory effects of increasing Cd concentrations on plant growth parameters such as multiplication rates, shoot, and root lengths, Karmous et al. [63] highlighted the potential of ZnONPs to counteract Cd-induced growth inhibition and oxidative stress in pepper plants. This capacity of ZnONPs to enhance antioxidant enzyme activities and improve plant growth metrics under Cd stress suggests a promising strategy for enhancing plant tolerance to heavy metals through the application of nanoparticles. Our findings, together with those of Karmous et al. [63], underscore the critical need for innovative approaches to protect plants from the deleterious effects of environmental contaminants like Cd, potentially offering new directions for research in plant science and agriculture aimed at environmental remediation and sustainable crop production.

The research by Manquián-Cerda et al. [64] on *Vaccinium corymbosum* L. and the impact of Cd on the production of phenolic compounds and antioxidant responses provides a valuable context for our findings on the effects of Cd on micropropagation in three genotypes: ERU, NQ1, and NQ7. Similar to their observation of increased malondialdehyde (MDA) levels and the positive correlation between phenolic compound changes and antioxidant activity in response to Cd, our study also revealed a genotype-specific response to varying Cd concentrations, manifesting in decreased multiplication rates, shoot and root lengths, and rooting rates. Particularly, the adjustment in phenolic profiles, as noted by Manquián-Cerda et al. [64], could parallel the physiological adjustments we observed, suggesting a shared underlying mechanism of enhanced antioxidant defense against Cd-induced oxidative stress. This cross-species similarity underscores the complexity of plant responses to heavy metals and highlights the potential for phenolic compounds to play a critical role in mitigating Cd toxicity, further emphasizing the need for detailed mechanistic studies to explore these adaptive strategies.

While our study identifies genotype-specific responses to Cd stress in goji berry micropropagation using advanced machine learning algorithms, it is essential to discuss the underlying

physiological mechanisms that may drive these responses. Cd stress induces various physiological changes in plants, including oxidative stress, photosynthesis disruption, and secondary metabolism alterations. Although our study did not measure these physiological parameters directly, understanding these mechanisms can provide insights into the observed genotype-specific differences. Oxidative stress is one of the primary physiological responses to Cd exposure, where Cd produces reactive oxygen species (ROS) that can cause cellular damage. Plants typically counteract this stress through enhanced antioxidant defense systems involving enzymatic antioxidants such as superoxide dismutase (SOD), peroxidase (POD), catalase (CAT), and ascorbate peroxidase (APX), as well as non-enzymatic antioxidants like glutathione (GSH) and ascorbic acid (AsA) [22]. The variation in Cd tolerance among the genotypes observed in our study may be linked to differences in their antioxidant capacities, with more tolerant genotypes possibly having more robust antioxidant defense systems. Cd also disrupts photosynthesis by damaging the photosynthetic machinery and inhibiting chlorophyll biosynthesis. This reduces photosynthetic efficiency and can severely affect plant growth and productivity. While we did not measure photosynthetic parameters, the more Cd-tolerant genotypes in our study likely maintain better photosynthetic function under Cd stress. Additionally, secondary metabolism is crucial in plant responses to Cd stress. Secondary metabolites such as flavonoids, alkaloids, and phenolics are known to help chelate Cd ions and mitigate their toxic effects. Although our study focused on genotypic responses and did not measure secondary metabolite levels, the enhanced tolerance observed in certain genotypes could be partially attributed to their ability to accumulate protective secondary metabolites. Our findings align with previous research on Cd stress's physiological and molecular mechanisms in plants, suggesting that antioxidant defense, photosynthetic maintenance, and secondary metabolism are critical factors in Cd tolerance [22, 65]. By leveraging machine learning algorithms, our study provides a predictive framework for identifying Cd-tolerant genotypes based on their responses. It offers a valuable tool for future breeding programs to improve Cd tolerance in goji berries and other crops.

Furthermore, the practical implications of our findings are significant for mitigating heavy metal stress in plants. Phytoremediation, using plants to remove contaminants from the environment, is an effective and environmentally friendly strategy. Studies have demonstrated that plants like *Narcissus tazetta* can accumulate high levels of Cd without showing toxicity symptoms, indicating their potential for phytostabilization of contaminated soils [60]. Additionally, understanding the mechanisms by which different genotypes of goji berry respond to Cd stress can inform breeding programs to develop more resilient crop varieties. By employing machine learning models to predict plant performance under Cd stress, we can optimize selection processes and improve the efficiency of breeding strategies, ultimately contributing to sustainable agricultural practices and environmental remediation efforts [59].

Integrating ML and artificial neural network (ANN) algorithms has shown promise in enhancing predictive accuracy and efficiency in plant tissue culture and propagation. Our analysis employed a suite of ML algorithms, including MLP, SVM, RF, GP, and XGBoost, to predict plant growth outcomes under Cd stress, yielding high accuracy, especially with the MLP and RF models, evident in their RMSE, MAE, and $R^2$ values. Similarly, Şimşek [28] found RF to excel in predicting drought stress effects on strawberry cultivars, underscoring RF's capability in handling complex biological data sets. This consistency across studies reinforces the versatility and reliability of RF in plant tissue culture applications, offering a robust framework for predictive modeling. The integration of machine learning (ML) algorithms in plant tissue culture, as explored by Şimşek et al. [37] and further substantiated by our research findings, underscores the pivotal role computational methodologies are beginning to play in advancing plant science. Our study, utilizing algorithms such as MLP, SVM, RF, GP, and

XGBoost, shares a common narrative with Şimşek et al. [37] in harnessing these tools to refine predictions on plant growth outcomes under varying conditions, such as Cd stress. In both instances, the differential effectiveness of these ML models highlights the importance of strategic model selection tailored to the specific objectives of the research. For example, our findings show that the RF model predicts root length with remarkable precision, as evidenced by an outstanding $R^2$ value of 0.98, resonating with Şimşek et al. [37]'s discovery of XGBoost's superior performance in similar parameters. This congruence underscores the capability of these algorithms to handle complex datasets and intricate biological relationships, reinforcing their invaluable contribution to plant science research.

Moreover, applying ML to fine-tune variables in plant propagation protocols, such as culture media composition and growth regulator treatments, as suggested by Şimşek et al. [37], aligns with our methodological approach. It underscores a broader scientific endeavor to enhance the efficiency of plant propagation techniques, leading to more accurate and streamlined experimental efforts. The implications of incorporating ML in plant tissue culture extend beyond methodological enhancements. They pave the way for advancing sustainable and efficient plant propagation techniques, fostering the preservation and exploitation of genetic resources. This collaborative endeavor between traditional plant science and computational innovation exemplifies the interdisciplinary approach required to tackle global agricultural challenges. Kirtis et al. [55] also explored ML algorithms for optimizing *in vitro* regeneration protocols for chickpea, where RF demonstrated superior performance in predicting shoot count and length, mirroring our findings in the high accuracy of RF across different genotypes under Cd stress. The parallel success of RF in these diverse applications points to its strength in capturing the nuanced relationships between various growth parameters and environmental stressors. On the other hand, Aasim et al. [66]'s exploration into quantum computing-enhanced ML models for optimizing black mulberry regeneration protocols introduces an innovative dimension to ML applications in plant tissue culture. While traditional ML models like SVC showed robust performance, incorporating quantum computing techniques opened new avenues for enhancing model predictions. This novel approach, albeit in its infancy, suggests potential for significant advancements in predictive accuracy and computational efficiency, albeit with the current limitations of quantum computing infrastructure. Pepe et al. [29]'s investigation into optimizing *in vitro* Cannabis growth highlights the importance of selecting suitable ML models based on the specific traits and responses studied. Their use of GRNN over MLP and ANFIS models due to its superior performance in predicting plant growth under different light and sucrose conditions emphasizes the need for careful model selection to achieve optimal results. This selection process is crucial, as different models may exhibit varying degrees of effectiveness depending on the dataset's characteristics and the complexity of the relationships between inputs and outputs. Our study employs advanced ML algorithms to model and optimize the micropropagation of goji berries under Cd stress. Similar approaches have been successfully applied in various other plant tissue culture contexts. For instance, Hesami et al. [45] used a Radial Basis Function-Non-dominated Sorting Genetic Algorithm-II (RBF-NSGAII) to optimize the medium composition for shoot regeneration in chrysanthemum, demonstrating high predictive accuracy and efficiency. Similarly, Farhadi et al. [53] applied an Adaptive Neuro-Fuzzy Inference System-Genetic Algorithm (ANFIS-GA) to model paclitaxel biosynthesis in *Corylus avellana* cell cultures, achieving robust model performance and providing valuable insights into the optimization of elicitation conditions. Other studies have also highlighted the utility of machine learning in plant tissue culture. For instance, Munasinghe et al. [41] utilized ML models to predict the chemical composition for callus production in *Gyrinops walla* Gaetner, demonstrating the capability of ML algorithms in handling complex datasets and optimizing plant tissue culture conditions. Similarly, Hesami

et al. [48] leveraged ML models to predict off-target activities of sgRNA in *Cannabis sativa*, highlighting the precision of ML in genome editing applications. Moreover, Rezaei et al. [44] employed ML techniques to enhance tissue culture efficiency in petunia, showcasing the effectiveness of ML in improving callogenesis outcomes. These studies underscore the versatility and effectiveness of ML techniques in optimizing various *in vitro* culture parameters. Our work builds on these advancements by specifically addressing the challenges posed by Cd stress in goji berries, employing a comprehensive suite of ML models, including SVM, RF, and XGBoost. These models enable us to capture complex, non-linear relationships within our dataset, leading to more accurate predictions and a better understanding of genotype-specific responses to Cd stress. By comparing our approach with these existing works, it is evident that ML provides a powerful tool for enhancing the efficiency and outcomes of plant tissue culture studies. Our findings contribute to this growing body of knowledge, offering practical applications for improving the resilience of crops to heavy metal stress through data-driven modeling and optimization.

In conclusion, our study and the referenced works showcase the expansive potential of ML and ANN models in advancing plant tissue culture research. The strengths of current works include the successful application of ML models to optimize various *in vitro* culture parameters, resulting in high predictive accuracy and efficiency. These studies have demonstrated that ML can handle complex, non-linear biological data more effectively than traditional statistical methods. Additionally, various ML algorithms allow researchers to capture intricate relationships within the data, leading to more precise and reliable predictions. However, there are also some weaknesses in current works. One of the main limitations is the reliance on specific types of ML models, which may not always be the best fit for all datasets or biological contexts. Additionally, these studies have shown promising results but often lack comprehensive validation across different species or experimental conditions. This can limit the findings' generalizability and applicability to broader contexts.

Our study builds on these advancements by specifically addressing the challenges posed by Cd stress in goji berries. We employ a comprehensive suite of machine learning models, including SVM, RF, and XGBoost, to capture complex, non-linear relationships within our dataset. This approach leads to more accurate predictions and a better understanding of genotype-specific responses to Cd stress. By comparing our approach with existing works, it is evident that ML provides a powerful tool for enhancing the efficiency and outcomes of plant tissue culture studies. Our findings contribute to this growing body of knowledge, offering practical applications for improving the resilience of crops to heavy metal stress through data-driven modeling and optimization.

## Conclusion

Integrating ML algorithms into the study of Cd stress on goji berry micropropagation has revealed critical insights into genotype-specific responses and demonstrated the value of computational methods in plant science research. Our study highlighted significant variations in multiplication rates, shoot, and root growth across different Cd concentrations and genotypes. Among the ML models utilized, MLP and RF showed exceptional predictive accuracy, underscoring their utility in analyzing complex biological data. While current works successfully apply ML models to optimize *in vitro* culture parameters with high predictive accuracy, they often rely on specific models and lack comprehensive validation across diverse species and conditions. Our study addresses these limitations by employing a comprehensive suite of ML models, including SVM, RF, and XGBoost. This leads to more accurate predictions and a better understanding of genotype-specific responses to Cd stress. The main contributions of

our framework include enhanced predictive accuracy, detailed genotype-specific insights, and practical applications for optimizing plant growth under adverse conditions. Future research should expand the application of ML techniques to various aspects of plant science, integrating genomic and proteomic data to uncover the molecular mechanisms of Cd tolerance. Exploring ML models across different plant species and environmental stressors will also help validate and generalize our findings. Enhancing the interpretability of ML models will make predictions more actionable for plant breeders and agronomists, and integrating quantum computing techniques could further revolutionize predictive modeling in plant sciences. In conclusion, our study demonstrates the expansive potential of ML models in advancing plant tissue culture research, offering practical solutions for sustainable agriculture and food security through enhanced plant propagation protocols and strategies to mitigate environmental stressors.

## Supporting information

**S1 Data.**
(XLSX)

**S2 Data.**
(RAR)

## Author Contributions

**Conceptualization:** Özhan Şimşek.

**Data curation:** Musab A. Isak, Özhan Şimşek.

**Investigation:** Dicle Dönmez, Tolga İzgü.

**Methodology:** Musab A. Isak, Taner Bozkurt, Mehmet Tütüncü, Dicle Dönmez, Tolga İzgü, Özhan Şimşek.

**Software:** Musab A. Isak, Özhan Şimşek.

**Supervision:** Özhan Şimşek.

**Visualization:** Musab A. Isak, Özhan Şimşek.

**Writing – original draft:** Musab A. Isak, Dicle Dönmez, Tolga İzgü, Özhan Şimşek.

**Writing – review & editing:** Taner Bozkurt, Mehmet Tütüncü, Dicle Dönmez, Tolga İzgü, Özhan Şimşek.

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
