## [Decision Letter · Decision Letter 0]

17 May 2024

PONE-D-24-15016Leveraging machine learning to unravel the impact of cadmium stress on goji berry micropropagationPLOS ONE

Dear Dr. Şimşek,

Thank you for submitting your manuscript to PLOS ONE. After careful consideration, we feel that it has merit but does not fully meet PLOS ONE’s publication criteria as it currently stands. Therefore, we invite you to submit a revised version of the manuscript that addresses the points raised during the review process.

We look forward to receiving your revised manuscript.

Kind regards,

Mojtaba Kordrostami, Ph.D.

Academic Editor

PLOS ONE

Journal Requirements:

Reviewers' comments:

Reviewer's Responses to Questions

**Comments to the Author**

1. Is the manuscript technically sound, and do the data support the conclusions?

Reviewer #1: Partly

Reviewer #2: Yes

2. Has the statistical analysis been performed appropriately and rigorously? 

Reviewer #1: No

Reviewer #2: Yes

3. Have the authors made all data underlying the findings in their manuscript fully available?

Reviewer #1: No

Reviewer #2: Yes

4. Is the manuscript presented in an intelligible fashion and written in standard English?

Reviewer #1: No

Reviewer #2: Yes

5. Review Comments to the Author

Reviewer #1: While the study demonstrates a comprehensive approach to examining the effects of cadmium stress on Goji Berry micropropagation across multiple genotypes using various machine learning algorithms, it requires major revisions before publication. The paper lacks clarity regarding the methodology employed in the experiments, including the specific protocols for cadmium stress application and micropropagation techniques. Additionally, there is insufficient detail provided on how the machine learning models were trained and evaluated, hindering reproducibility and validity. Moreover, while the study identifies genotype-specific responses to cadmium stress, it fails to adequately discuss the underlying physiological mechanisms driving these responses, limiting the depth of interpretation. Furthermore, the practical implications of the findings for mitigating heavy metal stress in plants are mentioned but not sufficiently explored or substantiated with empirical evidence or real-world applications. Overall, significant improvements in experimental design, methodology transparency, data interpretation, and practical relevance are necessary to enhance the rigor and impact of this study.

Introduction

The transition between discussing cadmium stress in plants and the application of machine learning in plant tissue culture is abrupt and lacks a clear connection. Providing a smoother transition or integrating these topics more seamlessly would enhance the flow of the introduction and help readers understand the relevance of each section to the overall study.

While the introduction briefly mentions the application of machine learning in plant tissue culture, it does not provide sufficient context or justification for this approach. Explaining the potential benefits of using machine learning in the context of micropropagation studies and highlighting the limitations of traditional statistical methods would strengthen this aspect of the introduction.

For instance, it would be beneficial to mention that:

The accuracy of machine learning has been approved for modeling, prediction, and optimization of different in vitro culture systems such as sterilization (https://doi.org/10.3389/fpls.2019.00282; 10.1371/journal.pone.0285657), seed germination (https://doi.org/10.1016/j.indcrop.2021.113753;
https://doi.org/10.1016/j.indcrop.2022.114801), callogenesis (https://doi.org/10.1016/j.inpa.2019.12.001;
https://doi.org/10.1007/s00253-021-11375-y;
https://doi.org/10.1371/journal.pone.0292359;
https://doi.org/10.1371/journal.pone.0293754), shoot proliferation (https://doi.org/10.1038/s41598-019-54257-0;
https://doi.org/10.3389/fpls.2021.757869;
https://doi.org/10.1007/s11240-022-02255-y;
https://doi.org/10.3390/app10155370), somatic embryogenesis (https://doi.org/10.1007/s11627-017-9877-7;
https://doi.org/10.1007/s00253-020-10978-1), haploid production (https://doi.org/10.1007/s00709-019-01379-x;
https://doi.org/10.3390/molecules26072053), gene transformation (https://doi.org/10.3389/fpls.2021.695110;
https://doi.org/10.1371/journal.pone.0239901), indirect shoot regeneration (https://doi.org/10.1186/s12896-023-00796-4), root formation (https://doi.org/10.3390/f13122020;
https://doi.org/10.1038/s41598-018-27858-4), and secondary metabolite production (https://doi.org/10.1186/s13007-021-00714-9;
https://doi.org/10.1371/journal.pone.0237478) and other aspects of tissue culture (https://doi.org/10.1007/s00253-020-10888-2).

The introduction should clearly outline the objectives and significance of the study. While it introduces various topics related to goji berry cultivation, cadmium stress, and machine learning, it is essential to explicitly state how these topics converge to address the research question or hypothesis of the paper.

Modeling Procedure

The division of the dataset into training and testing subsets using a 10-fold cross-validation method is appropriate for evaluating predictive performance. However, additional details on how the dataset was partitioned, such as the ratio of training to testing data and any stratification methods employed, would strengthen the reproducibility and robustness of the analysis.

While the utilization of R programming and relevant packages (Caret and Kernlab) for implementing machine learning algorithms is suitable, providing a brief overview of the specific functions and parameters used within these packages would assist readers in understanding the modeling process.

It's essential to address any potential biases or limitations in the experimental design or modeling approach. For example, discussing any inherent variability in the dataset or potential confounding factors that may influence the outcomes would strengthen the credibility of the study.

Results

It is essential that the authors include standard deviation values for Tables 1, 2, 3, 4, and 5 to provide a measure of the variability within the data. This addition is crucial for assessing the reliability and consistency of the results presented in these tables.

In Table 6, the absence of performance criteria (R2, RMSE, and MAE) for both the training and testing sets diminishes the comprehensiveness of the analysis. Including these metrics for both sets would offer a more comprehensive evaluation of the predictive performance of the models.

The absence of images depicting the in vitro-grown plantlets and their responses to cadmium stress is a notable limitation in the study. Visual representations play a crucial role in elucidating experimental findings and enhancing the reader's understanding of the observed phenomena. Integrating images would provide valuable insights into the morphological changes and responses of the plantlets under different experimental conditions.

Reviewer #2: This paper aimed to leverage machine learning to unravel the impact of cadmium stress on goji berry micropropagation. The topic is interesting, but there are still some ways could be improved:

More details on the motivations and experimental results should be clarified in the Abstract and Introduction Section.

To make the reference list cover more related works and improve the readability of this manuscript, I suggest that authors refer to the following works, if available:

1.Machine learning-based prediction of lymph node metastasis among osteosarcoma patients.

2.Development of a Machine Learning-Based Predictive Model for Lung Metastasis in Patients with Ewing Sarcoma.

More comparisons with SOTA works should be included to verify the efficiency of the proposed work.

Please clarify the reason that using proposed methods in this work. Why not some other novel machine learning methods?

Please present more comparisons with currents machine learning works.

Please also describe the strengthen and weakness of current works and the main contributions of the proposed framework.

More conclusions and discussions on future works need to be included.

6. PLOS authors have the option to publish the peer review history of their article (what does this mean?). If published, this will include your full peer review and any attached files.

Reviewer #1: No

Reviewer #2: No

---

## [Author Response · Author response to Decision Letter 0]

21 May 2024

Thank you again, the Editor and the reviewers, for your valuable comments on our manuscript. Please find our rebuttal to each peer's comment in the attachment, including what edits were made to address them.

We hope these significant edits merit publication in PLOS ONE.

Sincerely,

Assoc. Prof. Dr. Özhan Şimşek

---

## [Decision Letter · Decision Letter 1]

24 May 2024

Leveraging machine learning to unravel the impact of cadmium stress on goji berry micropropagation

PONE-D-24-15016R1

Dear Dr. Şimşek,

We’re pleased to inform you that your manuscript has been judged scientifically suitable for publication and will be formally accepted for publication once it meets all outstanding technical requirements.

Kind regards,

Mojtaba Kordrostami, Ph.D.

Academic Editor

PLOS ONE

Additional Editor Comments (optional):

Reviewers' comments:

Reviewer's Responses to Questions

**Comments to the Author**

1. If the authors have adequately addressed your comments raised in a previous round of review and you feel that this manuscript is now acceptable for publication, you may indicate that here to bypass the “Comments to the Author” section, enter your conflict of interest statement in the “Confidential to Editor” section, and submit your "Accept" recommendation.

Reviewer #1: All comments have been addressed

Reviewer #2: (No Response)

2. Is the manuscript technically sound, and do the data support the conclusions?

Reviewer #1: Yes

Reviewer #2: (No Response)

3. Has the statistical analysis been performed appropriately and rigorously? 

Reviewer #1: Yes

Reviewer #2: (No Response)

4. Have the authors made all data underlying the findings in their manuscript fully available?

Reviewer #1: Yes

Reviewer #2: (No Response)

5. Is the manuscript presented in an intelligible fashion and written in standard English?

Reviewer #1: Yes

Reviewer #2: (No Response)

6. Review Comments to the Author

Reviewer #1: All the comments have been addressed. I think that the current version of the manuscript can be published in Plos One.

Reviewer #2: (No Response)

7. PLOS authors have the option to publish the peer review history of their article (what does this mean?). If published, this will include your full peer review and any attached files.

Reviewer #1: No

Reviewer #2: No

---

## [Editor Report · Acceptance letter]

4 Jun 2024

PONE-D-24-15016R1 

PLOS ONE

Dear Dr. Şimşek, 

I'm pleased to inform you that your manuscript has been deemed suitable for publication in PLOS ONE. Congratulations! Your manuscript is now being handed over to our production team.

Kind regards, 

on behalf of

Dr. Mojtaba Kordrostami 

Academic Editor

PLOS ONE